# Understanding the Role of Plasma Bullet Currents in Heating Skin to Mitigate Risks of Thermal Damage Caused by Low-Temperature Atmospheric-Pressure Plasma Jets

Shunya Hashimoto [1,†], Hideo Fukuhara [2,†], Endre J. Szili [3,*], Chiaki Kawada [2], Sung-Ha Hong [3], Yuta Matsumoto [4], Tatsuru Shirafuji [1,4], Masayuki Tsuda [5], Atsushi Kurabayashi [6], Mutsuo Furihata [6], Hiroshi Furuta [7], Akimitsu Hatta [7], Keiji Inoue [2,8] and Jun-Seok Oh [1,4,*]

1   Graduate School of Engineering, Osaka City University, Osaka 558-8585, Japan
2   Department of Urology, Kochi Medical School, Kochi University, Kochi 783-8505, Japan
3   Future Industries Institute, University of South Australia, Adelaide, SA 5095, Australia
4   Graduate School of Engineering, Osaka Metropolitan University, Osaka 558-8585, Japan
5   Division of Laboratory Animal Science, Life and Functional Materials, Kochi Medical School, Kochi University, Kochi 783-8505, Japan
6   Department of Pathology, Kochi Medical School, Kochi University, Kochi 783-8505, Japan
7   Department of Electronic & Photonic Systems Engineering, Kochi University of Technology, Kochi 782-8502, Japan
8   Center for Photodynamic Medicine, Kochi Medical School, Kochi University, Kochi 783-8505, Japan
*   Correspondence: endre.szili@unisa.edu.au (E.J.S.); jsoh@omu.ac.jp (J.-S.O.); Tel.: +81-6-6605-3088 (J.-S.O.)
†   These authors contributed equally to this work.

**Abstract:** Low-temperature atmospheric-pressure plasma jets are generally considered a safe medical technology with no significant long-term side effects in clinical studies reported to date. However, there are studies emerging that show plasma jets can cause significant side effects in the form of skin burns under certain conditions. Therefore, with a view of developing safer plasma treatment approaches, in this study we have set out to provide new insights into the cause of these skin burns and how to tailor plasma treatments to mitigate these effects. We discovered that joule heating by the plasma bullet currents is responsible for creating skin burns during helium plasma jet treatment of live mice. These burns can be mitigated by treating the mice at a further distance so that the visible plasma plume does not contact the skin. Under these treatment conditions we also show that the plasma jet treatment still retains its medically beneficial property of producing reactive oxygen species in vivo. Therefore, treatment distance is an important parameter for consideration when assessing the safety of medical plasma treatments.

**Keywords:** nonthermal atmospheric pressure plasma; helium plasma jet; bullet currents; joule heating; gas temperature; mouse skin; skin burns

## 1. Introduction

Non-thermal atmospheric-pressure plasma technology is being widely investigated for various biomedical applications including disinfection [1–3], wound healing [4–8], cancer therapy [9–14], gene transfection [15–19], bone regeneration [20], and immunotherapy [14,21–23]. All these effects are strongly linked to the plasma generating reactive oxygen and nitrogen species (RONS) including charged species [24,25] and highly reactive neutral species such as the hydroxyl radical (•OH), singlet oxygen ($O_2(^1\Delta_g)$), nitric oxide (NO), and atomic oxygen (O) [26,27]. At atmospheric pressure, the plasma density can be ~$10^{13}$ cm$^{-3}$, even though the ionization degree is low or moderate [28]. At this plasma density, highly reactive RONS can be efficiently generated through interaction between the plasma components (electrons, ions, metastable atoms, and high-energy photons) and humid ambient air comprising molecular oxygen ($O_2$), nitrogen ($N_2$), and water ($H_2O$) vapour. Some of the more

highly reactive RONS are quickly converted to less reactive molecules, of which some are classified as RONS (e.g., hydrogen peroxide, $H_2O_2$) and some are not, such as nitrite ($NO_2^-$). Both the highly reactive and less reactive RONS and other molecules produced by plasma can intervene biological processes and/or modify biomolecules that lead to beneficial outcomes in disease treatments [29–31]. An atmospheric-pressure plasma jet (APPJ) operated as a dielectric barrier discharge (DBD) is one of the simplest methods to generate RONS in ambient air. Noble gases such as argon (Ar) or helium (He) are commonly used to run APPJs because they can be operated at lower voltages, which helps reduce gas heating, making these plasma sources suitable for treating thermally sensitive material such as human skin. Sometimes molecular gases such as $O_2$ and $N_2$ or $H_2O$ vapour are added into the noble gas stream to enhance the production of RONS [32].

Remarkably, plasma can produce effects in disease treatment that penetrate to significant depths in the human body. This includes plasma treatments reducing the growth of millimetre-thick 3D tumours [13,14] and inducing systemic physiological responses such as stimulation of the immune system in cancer therapy [21–23]. Even though plasma itself cannot penetrate a physical barrier such as human skin, we have observed that plasma has the potential to produce RONS at millimetre depths in skin [13,33–38]. Interestingly, plasma generally appears to non-invasively produce RONS through biological barriers such as skin, i.e., without damaging the surface of barrier [39–41]. The ability of plasma to produce RONS at millimetre depths in biological tissue may stimulate a cascade of biological events that lead to deep and systemic effects important in disease treatment. However, this is not always the case, as the nature of the biological barrier regarding characteristics such as hydration and conductivity can change the plasma properties in a way that damages the biological target being treated. For example, it has been shown that localized heating from a He plasma jet, coupled with chemical modification from plasma produced RONS, can result in significant damage to mouse skin [42].

In our previous work, we have been developing a He plasma jet to treat cancer and have shown it can successfully shrink subcutaneous tumours in mice [13]. However, one side effect we have recently noticed under certain conditions is that the plasma treatment can also damage the mouse skin. For example, Figure 1a shows a millimetre-sized discoloured spot created on the back of a mouse following 15 min of helium plasma jet treatment of the skin directly above a subcutaneous tumour. The mouse was sacrificed 24 h after the plasma treatment, and the skin was collected for histological analysis. The cross section at the centre of the plasma-irradiated skin sample showed a scab above the epidermis, indicating the skin was wounded by the plasma treatment (Figure 1b).

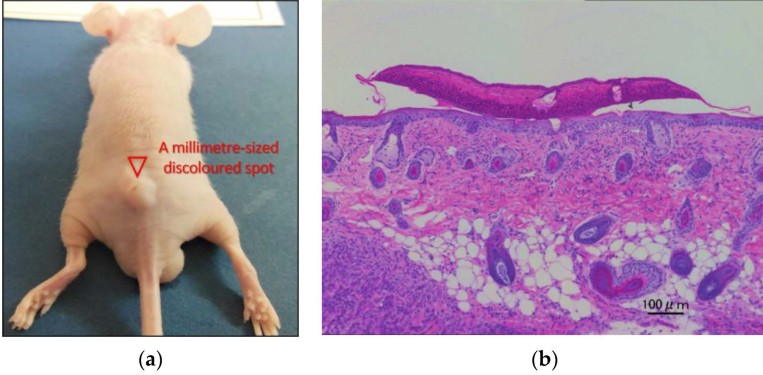

(a)                                                   (b)

**Figure 1.** (**a**) Photograph of a seven-week-old female BALB/c nu/nu mouse with a subcutaneous cancer tumour following 15 min helium plasma jet treatment. The triangle indicates a millimetre-sized discoloured spot created in the plasma-treated region on the skin above the subcutaneous tumour. (**b**) Histological cross-section of the discoloured region taken 24 h after the plasma treatment. The microscope image shows a scab, indicating a wound created in the discoloured region.

Therefore, this study aims to provide further insights into what plasma components damage skin. We also demonstrate how understanding the damaging plasma components can be used to prevent skin damage whilst still maintaining high efficiency in production of RONS in mice.

## 2. Materials and Methods

A low-temperature atmospheric-pressure helium microplasma jet was employed in this study [43]. It consisted of a 150 mm long glass tube tapered from an inner diameter of 4 mm to 680 μm at the nozzle. The glass tube (borosilicate, Pyrex, As One Co., Osaka, Japan) has a 15 mm long metallic external ring electrode wound onto the glass tube at 50 mm from the end of the nozzle. He gas (99.98%, industrial grade) was fed into the glass tube with a fixed gas flow rate of 2.0 L/min. A capillary dielectric barrier discharge (DBD) microplasma was generated using a sinusoidal high voltage of 10 kV$_{\text{p-p}}$ (peak-to-peak) applied to the external electrode at a fixed frequency of 33 kHz. This particular plasma jet configuration and these operational conditions were chosen because they produced a relatively long plasma jet (exiting the nozzle) with a plume length of 12 mm and a gas temperature of 40 °C, as estimated by optical emission analysis of the N$_2$ s-positive system [44,45]. Plasma treatments were carried out using two distances between the nozzle and mouse skin: 10 mm, where the plasma jet contacted the mouse skin (contact plasma, Figure 2a), and 12 mm, where the plasma jet was not in contact with the mouse skin (non-contact plasma, Figure 2b). Treatments were performed for up to 15 min because in our previous study we found this was the minimum time required to obtain effective cancer treatment [13].

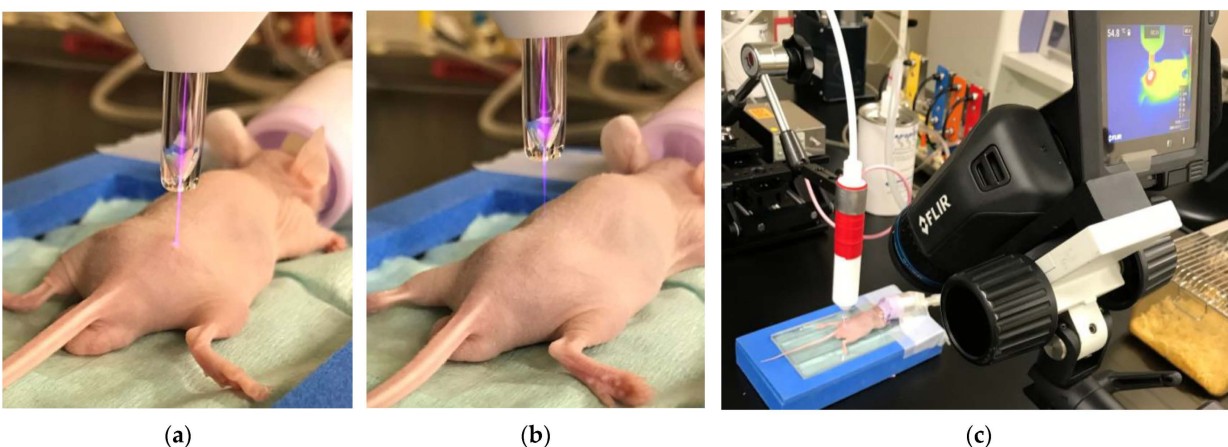

(**a**) (**b**) (**c**)

**Figure 2.** Photographs show: (**a**) low-temperature atmospheric-pressure helium microplasma jet treating a live mouse as an example of contact plasma treatment; (**b**) non-contact plasma treatment; and (**c**) the complete experimental set-up with the position of the thermography camera to measure skin temperature.

The temperature of the mouse skin during plasma treatment was directly measured by a thermography camera (FLIR T530, Wilsonville, OR, USA) as shown in Figure 2c during and after plasma treatments. This method was chosen because it enabled us to obtain non-invasive measurements of the mouse skin temperature in real time and without perturbing the plasma treatments. For comparison, mouse skin temperature was also measured in controls of untreated mice and mice treated with He gas flow only (i.e., with plasma off). Mouse skin temperatures for contact plasma treatment were also simulated with the COMSOL Multiphysics software (COMSOL, Inc., Burlington, MA, USA) to predict the radial temperature distribution across the skin.

For the experiment, 7-week-old male BALB/c nu/nu mice were housed in plastic cages with stainless steel grid tops in an air-conditioned room with a 14 h light–10 h dark cycle maintained at a temperature of 23 ± 2 °C and provided with water and food ad libitum in the institute for animal experiments of Kochi Medical School. Animal experiments

were conducted according to the institutional guidelines and regulations. The animal experimental protocol was approved by the Institutional Animal Care and Use Committee of Kochi University under Permit Numbers J-16, L-6, and M-27. Mice skin for separate experiments in this study was harvested from live mice under the same animal ethics standards. For in vivo detection of internal ROS, each mouse was exposed to the He plasma jet or He gas flow for 15 min. After treatment, the mice immediately received an intraperitoneal injection of 500 μL of luminol solution prepared in sterile saline. Reactive oxygen species (ROS) produced by the He plasma jet treatment react with luminol to form a luminescent product underneath the skin inside the live mouse. The luminescent product can be mapped throughout the mouse via luminescence imaging. Luminescence imaging was carried out 10 min after treatment (Xenogen IVIS Spectrum 200, PerkinElmer, Waltham, MA, USA) [13].

The plasma bullet current was determined via measuring the current flow through a metal (copper) collector plate with a diameter of 10 mm connected to the ground via an electrical wire as shown later section in results and discussion [24]. The ground wire was inserted through a current monitor (Pearson 2877, London, UK) to measure the current during plasma treatments. The current was measured directly through the collector plate or indirectly by covering the plate with mouse skin sections of 30 mm × 30 mm. This method enabled us to estimate the amount of current flowing through the skin during plasma jet treatments of the live mice.

## 3. Results and Discussion

### 3.1. Mouse Skin Temperature

To understand the role of the plasma components in heating mouse skin, the surface skin temperature of live mice was recorded in the 15th min of the plasma treatment. Firstly, Figure 3a shows the mouse has a temperature of approximately 30 °C uniformly distributed across its surface. When exposed to He gas flow only (i.e., without plasma ignition), the surface skin temperature dropped to approximately 27 °C over a circular area of diameter 20 mm, as seen in Figure 3b. This cooling effect is because the He gas temperature is lower than the ambient air because it cools as it expands when it exits the compressed bottle. The He cools the mouse skin over an area much larger than the 680 μm diameter of the nozzle, indicating that the gas propagates across the skin as it hits its surface. The temperature of remote parts of the mouse skin not directly exposed to the He flow remained constant at 30 °C. Contact plasma treatment increased the mouse skin temperature to an average of 40 °C with a hotspot of 59 °C in the region directly exposed to the plasma jet (Figure 3c). The increase in skin temperature expanded over a circular area with a diameter of 12 mm, which was much larger than the nozzle diameter but smaller than the diameter of 20 mm cooled by the He gas flow only in Figure 3b. The smaller area heated by the plasma jet compared to that cooled by the He gas flow could be due to the gas temperature and the charged species in the plasma jet directing the gas flow to the mouse skin due to ion momentum transfer to the neutral gas [46,47]. Another explanation is that the plasma itself may be heating the mouse skin directly (as opposed to indirectly by heating the background gas). To test this, the temperature of the mouse skin was measured during non-contact plasma treatment. Non-contact plasma treatment resulted in only a negligible increase in average skin temperature to 40 °C and no hotspots (Figure 3d). The change in skin temperature covered an area similar to that produced by the contact plasma treatment due to spread of the heated He gas across the mouse skin. The higher temperature observed on the mouse skin during the contact plasma treatment is therefore not only attributed to the heated He gas but also to other components in the plasma itself.

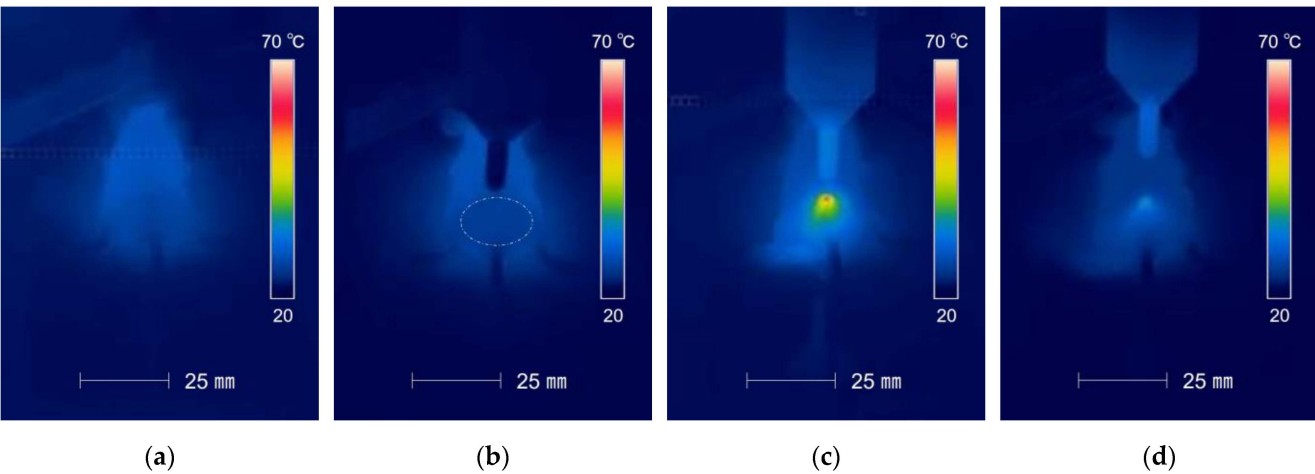

**Figure 3.** Thermal images taken on live mice (**a**) before treatment, (**b**) during He gas flow treatment at 15 min, (**c**) during contact plasma treatment at 15 min, and (**d**) during non-contact plasma treatment at 15 min.

The maximum temperatures produced on the mouse skin by the treatments described above are plotted in Figure 4. The graph shows that contact plasma treatment significantly elevates the skin temperature to 59 °C, which is high enough to injure skin, whereas the maximum skin temperature remains below 40 °C during non-contact plasma treatment, which is not high enough to injure skin. The temporal change in skin temperature was also monitored during the 15 min of plasma treatment and 3 min after the treatment was finished (with both plasma and He gas flow off). Figure 4a shows that the mouse skin temperature remained relatively constant at 38 °C over a measurement time period of 15 min. He gas treatment quickly decreased the skin temperature to 28 °C, after which it remained constant over the 15 min treatment.

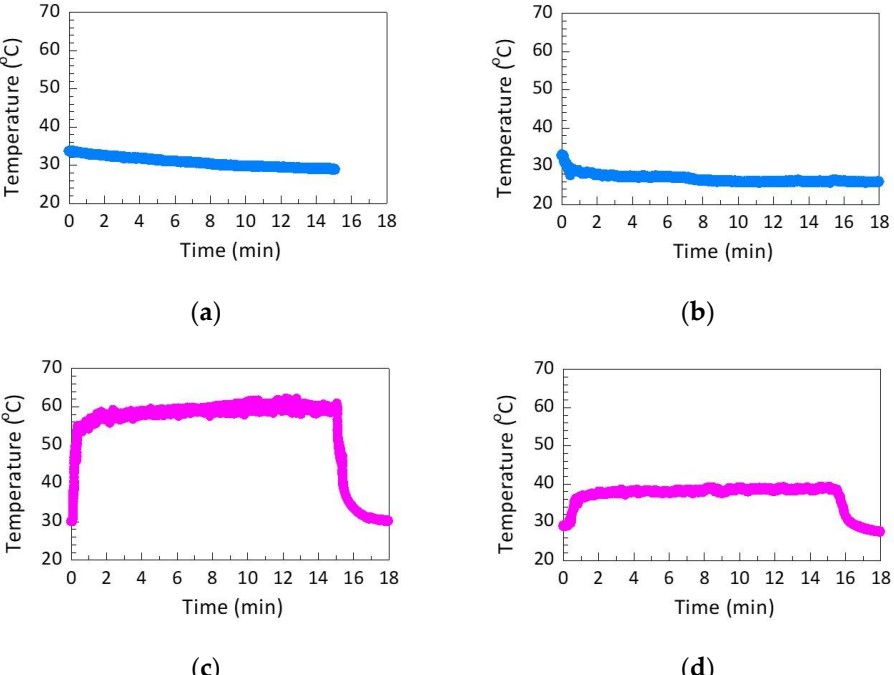

**Figure 4.** Temporal changes in the mouse skin temperature (**a**) for untreated mice, (**b**) during He gas flow treatment, (**c**) during contact plasma treatment, and (**d**) during non-contact plasma treatment. Plasma treatments were performed for 15 min with temperature recorded for up to 18 min (i.e., up to 3 min post treatment). All data are representative of averages of triplicate measurements.

For contact plasma treatment, the mouse skin temperature rapidly increased within the first 30 s to a maximum of 59 °C and then remained constant at this temperature over the 15 min treatment (Figure 4c). After the treatment was finished and the plasma and He gas flow were switched off, the mouse skin temperature quickly decreased back to its original temperature of 30 °C. Non-contact plasma treatment also quickly increased the mouse skin temperature but to a temperature below 40 °C, which would not injure the mouse skin (Figure 4d). Similar to contact plasma treatment, the elevated mouse skin temperature remained constant over the course of the 15 min non-contact plasma treatment and then returned to baseline after the treatment was completed. Overall, the data show that contact plasma treatment results in a quick and significant increase in temperature that can injure mouse skin and that the temperature remains elevated over the course of the treatment. Figure 5 shows the average temperatures during the time of plasma treatment. The cumulative thermal damage to the mouse skin over the course of prolonged plasma treatment (e.g., 15 min) may explain the skin wound created by the plasma treatment in Figure 1.

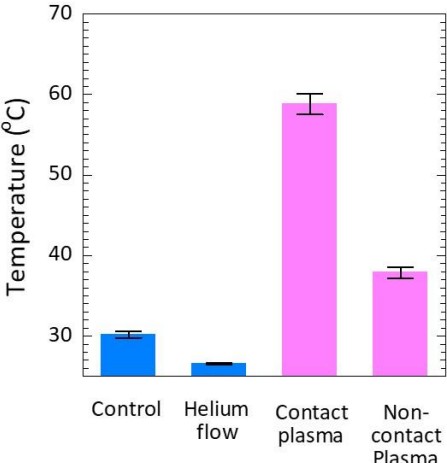

**Figure 5.** Maximum temperature of mouse skin for untreated mice during He gas flow treatment at 15 min, during contact plasma treatment at 15 min, and during non-contact plasma treatment at 15 min. All data are representative of averages of triplicate measurements.

### 3.2. Correlation between Plasma Bullet Current and Mouse Skin Temperature

This section sets out to determine if the plasma bullets have an important role in heating mouse skin during the contact plasma jet treatment. To determine this, this study analysed the correlation between the plasma bullet current produced through mouse skin and the rise in mouse skin temperature induced by the plasma treatments. It is known that the charged (active) species within a plasma jet are contained in a succession of hypersonic propagating plasma bullets [48,49]. Previously, using ambient mass spectrometry analysis of the charged plasma species, we have shown that positive and negative ionic species are detected along the length of the plasma plume, i.e., the visible part of the plasma jet [24]. However, the intensity of the ionic species rapidly declines at the tip of plasma plume and then drops by three orders of magnitude at distances remote from the plasma plume. In another study where we investigated the plasma bullet current, we observed that the net positive and negative charges in the plasma bullet decreased as functions of distance from the end of the glass nozzle of the plasma jet assembly [46]. Net cha-rges dropped by two orders of magnitude at the tip of plasma plume compared to the charges within the centre of the plasma plume. These results suggest that charged species in the plasma plume may have a major role in increasing the mouse skin temperature.

Therefore, we developed a simple experiment to test the hypothesis that the plasma bullet current increases the mouse skin temperature. The experimental set-up to measure currents transmitted through the mouse skin is shown in Figure 6. This was achieved with the aid of a collector plate connected to the ground wire. The current flowing in this circuit

was measured with a current monitor. At the same time, the temperature was also measured by a thermal imaging camera. Figure 7a shows the current waveforms recorded through the mouse skin during contact plasma treatment with and without the mouse skin being on top of the collector plate. The current waveforms are typical of plasma bullets produced in capillary DBDs, i.e., representative of sharp positive and broad negative current pulses. The amplitude of the positive current pulse was measured to be 3.8 mA without mouse skin, while it was 2 mA with mouse skin. The positive current pulse with mouse skin appeared a few microseconds earlier than without mouse skin. The negative current pulse had an amplitude of $-0.7$ mA with and without mouse skin. These results indicate that the plasma bullet currents can penetrate through mouse skin of <1 mm thickness.

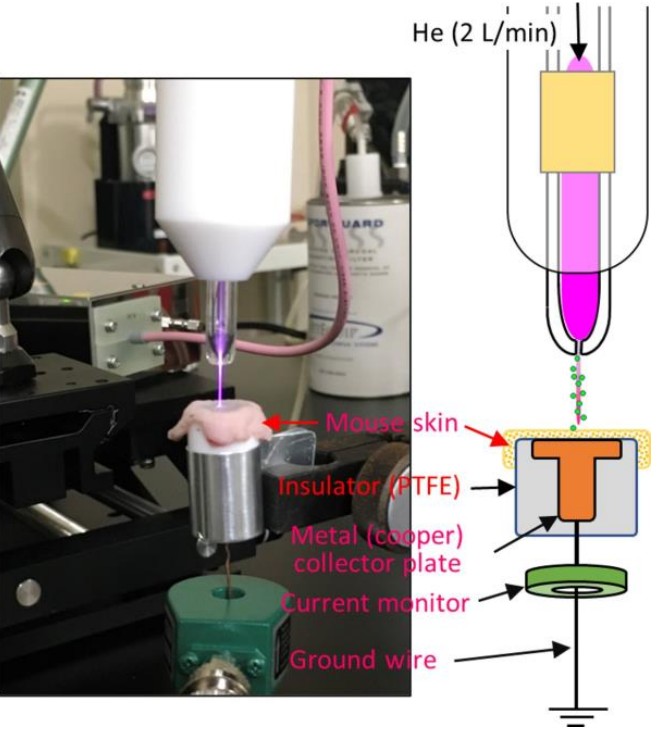

**Figure 6.** Photograph and schematic of the experimental set-up to measure the current produced through mouse skin by the plasma bullets.

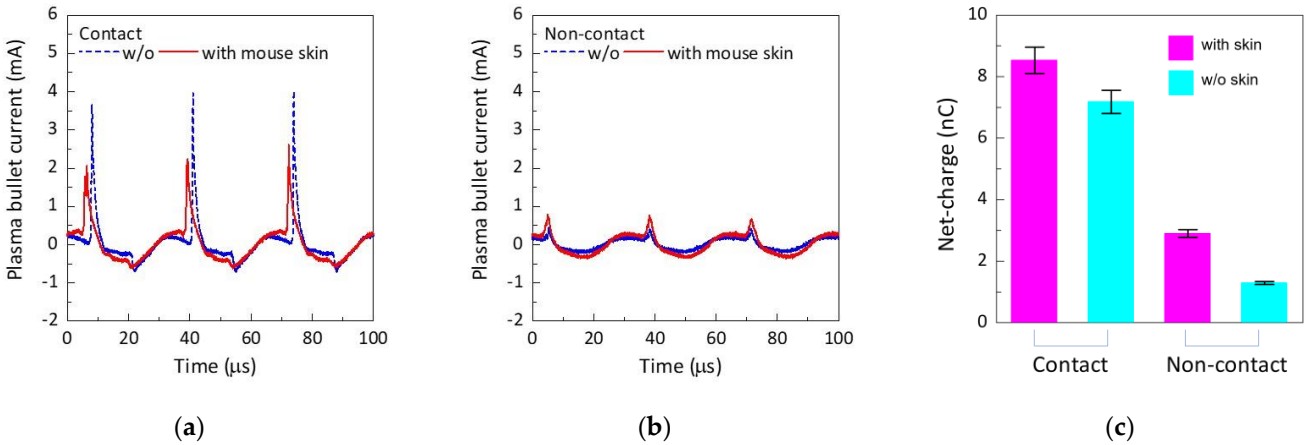

**Figure 7.** Current waveforms (averaged) recorded through the mouse skin during (**a**) contact plasma treatment and (**b**) non-contact plasma treatment. The total net charges measured through the mouse skin for contact and non-contact plasma treatments are shown in (**c**). All data are representative of averages of triplicate measurements.

For non-contact plasma treatment, the current peaks were significantly lower, as shown in Figure 7b. The positive current peaks were 0.4 mA without mouse skin and 0.7 mA with mouse skin. The negative current peaks could not be measured, presumably because these values were below the detection limit (0.01 mA) of the current monitor. Overall, the results show currents transmitted through mouse skin are significantly lower when the plasma plume does not contact the mouse skin.

The total net charge ($Q_n$) of the positive bullet current pulse $I_B^+$ and the negative bullet current pulse $I_B^-$ through the mouse skin was calculated, according to equation below:

$$Q_n = \int_0^T |I_B^\pm| \, dt, \tag{1}$$

where $Q_n$ is the net charge and $T$ is the time for a period. For contact plasma treatments the total net charge was calculated to be $8.53 \pm 0.43$ nC, which was significantly higher compared to the $2.90 \pm 0.13$ nC measured during non-contact plasma treatment, as shown in Figure 7c. Therefore, it is highly likely that the plasma bullet currents produced by the He plasma jet in this study heat mouse skin by transferring electric current through the skin in a process called joule heating.

Figure 8 shows the correlation between the net charge recorded through the mouse skin and the change in skin temperature during the 15 min of plasma treatments and 3 min after the treatments were completed. Once again, contact plasma treatment quickly increased the temperature on the mouse skin but to a higher temperature of 80 °C compared to the maximum temperature of 59 °C recorded during treatment of the live mice (Figure 8b). The higher temperature in the mouse skin section is due to the lack of the thermoregulatory system used to control temperature in mice. Mouse skin temperature remained constant over the course of the plasma treatment and then quickly declined to baseline after the treatment was completed. Non-contact plasma treatment still did not considerably heat the mouse skin, with the maximum temperature reaching 40 °C (Figure 8b).

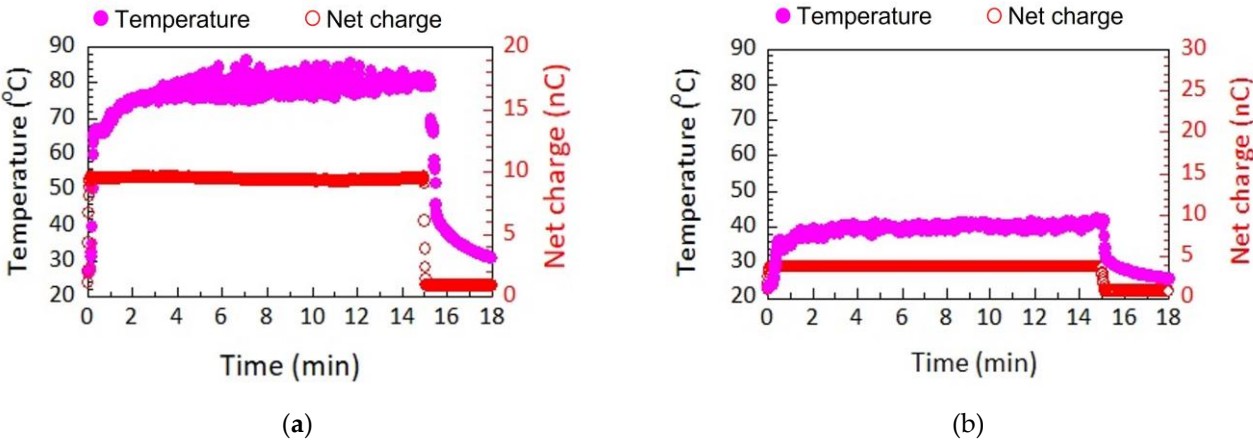

**Figure 8.** Correlation between temporal changes in mouse skin temperature and net charge through mouse skin during (**a**) contact and (**b**) non-contact plasma treatments. Plasma treatments were performed for 15 min, with temperature recorded for up to 18 min (i.e., up to 3 min post-treatment). All data are representative of averages of triplicate measurements.

The temporal changes in the net charge for both contact and non-contact plasma treatments were plotted on the same graphs along with the temporal temperature changes in Figure 8. For both contact and non-contact plasma treatments it is clearly seen that the profile changes in the net charges recorded through the mouse skin directly correlate with the temperature changes recorded on the mouse skin.

The results so far indicate that the plasma bullet current may cause significant heating of the mouse skin. This heating is also non-uniform, with potentially damaging hotspots seen in Figure 3c during contact plasma treatment. To interrogate the hotspots, we mapped

the spatial distribution of the mouse skin temperature along the radial centre shown in Figure 9a for contact plasma treatment and Figure 9b for non-contact plasma treatment at different treatment times of 0, 5, 10, and 15 min. Contact plasma treatment produced a hotspot across a narrow region that did not change as a function of treatment time. Non-contact plasma treatment produced a significantly cooler and broader hotspot.

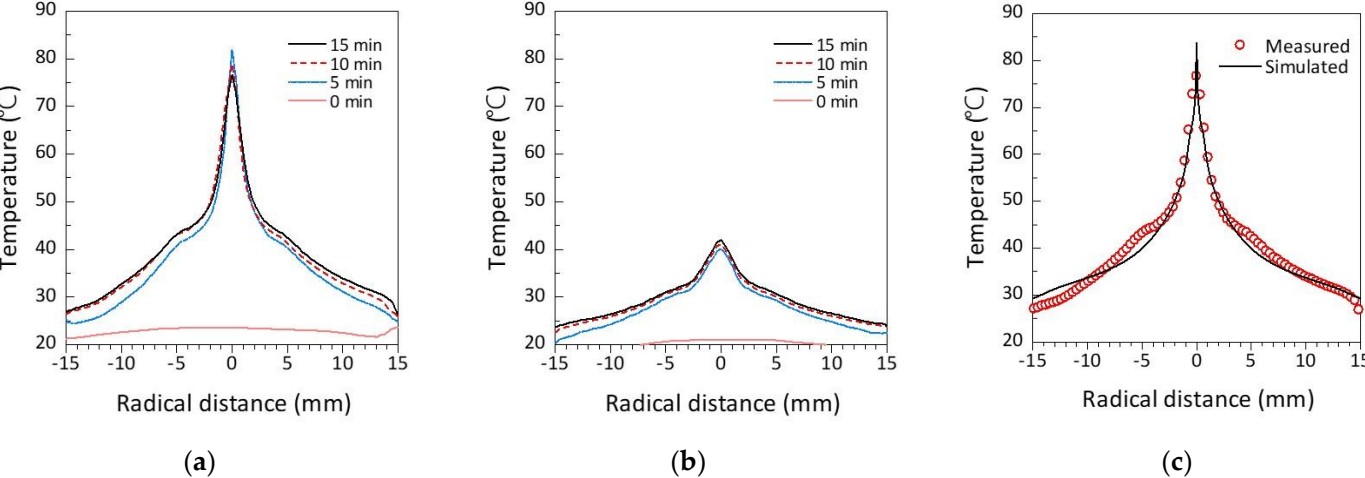

**Figure 9.** Radial temperature distribution across mouse skin during (**a**) contact plasma treatment, and (**b**) non-contact plasma treatment. In (**c**), the measured radial temperature distribution across mouse skin for contact plasma treatment is compared to a COMSOL Multiphysics simulated temperature profile.

To theoretically validate the experimental results and confirm if the plasma bullet currents are indeed responsible for heating the mouse skin, we performed a simulation using COMSOL Multiphysics of the contact plasma treatment to predict the radial temperature distribution across the mouse skin. We used an input power of 0.16 W and a resistance of mouse skin of 0.71 MΩ, which was estimated from the plasma bullet current measurements. The input power was used as the heat source. The laminar flow interface [50,51], heat transfer in fluids interface, and non-isothermal flow were used to solve the radial temperature distribution. As seen in Figure 9c, the simulation closely followed the experimental results. Therefore, we can conclude that the heating of the mouse is indeed mainly caused by joule heating from the plasma bullet current.

### 3.3. In Vivo Plasma Production of ROS

It was shown in this study that non-contact plasma treatment does not thermally damage mouse skin. Therefore, this may be a safer option to treat mice compared to contact plasma treatment. However, to understand if non-contact plasma treatment can be useful for disease treatment, we also need to understand if the treatment can also produce RONS in mice at similar levels to those that can be achieved via contact plasma treatment. Therefore, we performed a final experiment to image the spatial distribution of ROS inside live mice after contact and non-contact plasma treatments. The experiment involved injecting mice with luminol, which reacts with ROS to form a bioluminescent product that can be imaged. Figure 10a,b show the luminescence produced in the mice by the contact and non-contact plasma treatments, respectively. Both treatments resulted in a prominent increase in luminescence, which suggests that both treatments produce ROS inside the live mice. The luminescence intensity counts were plotted in Figure 10c. The graph clearly shows that non-contact plasma treatment produces a same level of ROS in vivo as compared to contact plasma treatment.

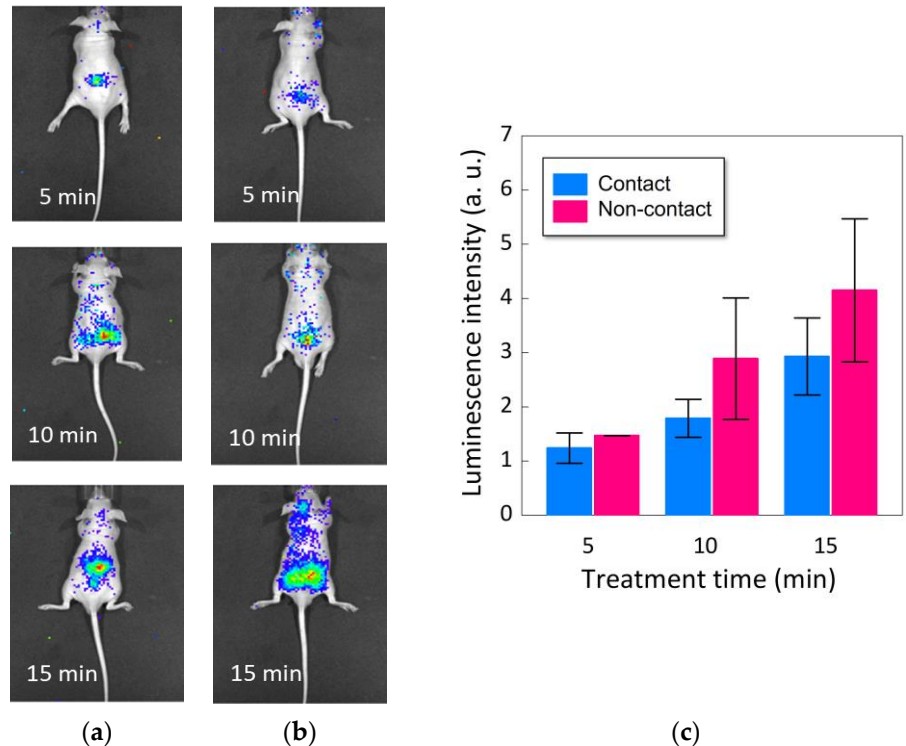

**Figure 10.** In vivo detection of ROS (luminescence resulting from reaction of luminol with ROS) following (**a**) contact and (**b**) non-contact plasma treatments. The averaged luminescence intensity counts (from *n* = 5 mice) are plotted in (**c**).

## 4. Conclusions

This study shows that a low-temperature atmospheric-pressure helium plasma jet can damage mouse skin during treatment. This damage is in the form of a thermal burn and is caused by joule heating from the plasma bullet current, which was shown experimentally and confirmed with a COMSOL Multiphysics simulation. However, the skin damage can be prevented by treating mice at a further distance so that the visible plasma plume does not contact the skin. At the longer distance it is still possible to produce reactive oxygen species in the mice with the same level of efficiency achieved by the closer-distance treatment. Therefore, this study provides new insights into how low-temperature atmospheric-pressure plasma jets damage skin and a solution to prevent this damage, which is useful knowledge when developing plasma jets for medical applications.

**Author Contributions:** This study was conceptualized by H.F. (Hideo Fukuhara), E.J.S., K.I., M.F., H.F. (Hiroshi Furuta), A.H. and J.-S.O. Methodology and investigations were performed by S.H., H.F. (Hideo Fukuhara), C.K., Y.M., T.S., M.T., A.K. and J.-S.O. Draft was written by S.H., H.F. (Hideo Fukuhara), E.J.S. and J.-S.O. and revised by S.-H.H., E.J.S. and J.-S.O. All authors have read and agreed to the published version of the manuscript.

**Funding:** This work supported by JSPS KAKENHI Numbers JP19K18564 and JP22K09505. E.J.S. acknowledges the support from the Australian Research Council Future Fellowship FT190100263 and the National Health Medical Research Council Ideas Grant 2002510. J.-S.O. acknowledges the support from Osaka City University Strategic Research Funds (Priority Research) in FY2019 and financial support by the BioMedical Engineering Center (BMEC).

**Institutional Review Board Statement:** Not applicable.

**Informed Consent Statement:** Not applicable.

**Data Availability Statement:** Not applicable.

**Conflicts of Interest:** The authors declare no conflict of interest.

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
