# Peer review of "Understanding the Role of Plasma Bullet Currents in Heating Skin to Mitigate Risks of Thermal Damage Caused by Low-Temperature Atmospheric-Pressure Plasma Jets"

_plasma, doi:10.3390/plasma6010009_

Round 1
Reviewer 1 Report
This article describes the plasma treatment on mice. It was concluded that contact plasma will damage the skin, and non-contact plasma doesn’t damage skin by providing quantitative measurement. I have a few minor comments.
- The full name of the abbreviation should be given when the first time it is used. For example, DBD in line 102.
- The caption of figure 3 doesn’t have the description of (b).
- This article is straight forward. The authors should make it easier to follow for a broader audience. The authors can give an example when describing the method. For example, it is very similar to ignite a gas stove or a gas burner, and the difference is …… rather than just using the technical term.
Author Response
Thank you for the valuable reviewers’ comments. We here reply to the comments. In addition, we found incorrect reference numbers in the manuscript. All the amendments are highlighted in the revised manuscript.
Please find attached.

Reviewer 2 Report
The paper presents experimental results on how a low-temperature atmospheric-pressure helium plasma jet affect mouse skin during contact/non-contact plasma treatment. The paper show that skin burns are caused by joule heating from the plasma bullet current and that the damage is mitigated by treating the mice at a further distance. It is also shown that reactive oxygen species in the mice are produced in non-contact plasma treatment with the same level of efficiency as the closer distance treatment.
The results and discussions are clearly presented at a satisfactory level, and I would like to drop a few suggestions for improving the paper.
- line 196 in page 6 / line 292 in page 9
The topics change between the paragraphs, so that I would suggest dividing section 3 into three subsections and providing somehow an outline at the beginning of the section.
- Figure 7(c) in page 7
In my environment, the background color of Fig. 7(c) is black, and it is hard to see the figure. Please correct it.
Author Response

(The authors gave the same response as above.)
